# Multi-Omics Analyses to Identify FCGBP as a Potential Predictor in Head and Neck Squamous Cell Carcinoma

**DOI:** 10.3390/diagnostics12051178

**Published:** 2022-05-09

**Authors:** Yu-Hsuan Lin, Yi-Fang Yang, Yow-Ling Shiue

**Affiliations:** 1Institute of Biomedical Sciences, National Sun Yat-sen University, Kaohsiung 804, Taiwan; lucaslinyh@gmail.com; 2Department of Otolaryngology, Head and Neck Surgery, Kaohsiung Veterans General Hospital, Kaohsiung 813, Taiwan; 3School of Medicine, National Yang Ming Chiao Tung University, Taipei 112, Taiwan; 4School of Medicine, Chung Shan Medical University, Taichung 402, Taiwan; 5Department of Medical Education and Research, Kaohsiung Veterans General Hospital, Kaohsiung 813, Taiwan; yvonne845040@gmail.com; 6Institute of Precision Medicine, National Sun Yat-sen University, Kaohsiung 804, Taiwan

**Keywords:** head neck squamous cell carcinoma, IgG Fc binding protein, FCGBP, prognosis, tumor microenvironment, immune

## Abstract

(**Purpose**) Previous studies have pointed out the significance of IgG Fc binding protein (FCGBP) in carcinogenesis, cancer progression, and tumor immunity in certain malignancies. However, its prognostic values, molecular interaction, and immune characteristics in the head and neck squamous cell carcinoma (HNSC) remained unclear. (**Methods**) To evaluate the potential role of the *FCGBP* gene, we used GEPIA2 and UALCAN platforms to explore the differential levels, survivals, and genetic alteration through cBioPortal (based on The Cancer Genome Atlas dataset). STRING, GeneMania, and TIMER2.0 identified the interacting networks. LinkedOmics performed Gene enrichment analysis, and TISIDB and TIMER2.0 evaluated the role of *FCGBP* in the tumor microenvironment. (**Results**) The expression level of *FCGBP* is lower in cancer tissues. A high *FCGBP* level is significantly associated with better overall- and disease-specific-survivals, regardless of human papillomavirus infection. Low *FCGBP* levels correlated to a higher *tumor protein p53* (*TP53)* mutation rate (*p* = 0.018). *FCGBP* alteration significantly co-occurred with that of *TP53* (*q =* 0.037). Interacting networks revealed a significant association between FGFBP and trefoil factor 3 (TFF3), a novel prognostic marker in various cancers, at transcriptional and translational levels. Enrichment analyses identified that the top gene sets predominantly related to immune and inflammatory responses. Further investigation found that the *FCGBP* mRNA level positively correlated to the infiltration rates of B cells, Th17/CD8+ T lymphocytes, T helper follicular cells, mast cells, and expression levels of various immune molecules and immune checkpoints in HNSC. (**Conclusions**) We found that the *FCGBP* mRNA level negatively correlated to *TP53* mutation status while positively correlated to the *TFF3* level. Additionally, FCGBP may regulate the tumor microenvironment. These findings support the *FCGBP* as a potential biomarker to estimate HNSC prognoses.

## 1. Introduction

Head and neck squamous cell carcinoma (HNSC) ranks the 7th among cancer-related deaths, with over 890,000 new cases annually worldwide and mortality of approximately 50% [1]. Leading causes of HNSC include alcohol consumption, smoking, and high-risk human papillomavirus (HPV) infection [2]. Although the growing trend in HNSC may vary with ethnicities, the annual incidence of HPV-related HNSC is consistently increasing [2,3]. Cancer behaviors are distinct between HPV-related and HPV-unrelated HNSCs. Overexpression of cyclin-dependent kinase inhibitor 2A (CDKN2A/p16), downregulation of tumor protein p53 (TP53), and inactivation of RB transcription corepressor 1 (RB1), have been characterized in HPV-related HNSC [4]. The underlying molecular mechanisms were identified as HPV oncoprotein E6 ubiquitinate TP53, another oncoprotein E7 eliminated RB1 through proteasomal degradation of RB1 to facilitate cell cycle progression, thereby transactivation of the *CDKN2A* via E2F proteins releasing from RB1/E2F family transcription factors [4]. In terms of clinical manifestations, HPV-related HNSC significantly correlated with favorable treatment responses after radiotherapy and chemotherapy, leading to a better prognosis [5]. Nevertheless, the overall survival (OS) of HNSC did not improve considerably despite multidisciplinary advances [5]. Consequently, it is essential to explore novel biomarkers for effective therapeutic strategies.

The *FCGBP* gene encoded a cysteine-rich glycoprotein comprising approximately 5400 amino acid residues, which may bind the Fc portion of the IgG molecule [6]. The mucous epithelia of various organs, including the intestine and colon, gallbladder, salivary gland(s), and cervix uterus, may synthesize FCGBP, thus becoming a constituent of many body fluids [6,7]. Physiologically, FCGBP protects the cells against infections from microorganisms [8]. One speculated mechanism is that FCGBP modulates the innate mucosal immunity to defense by binding the trefoil factor 3 (TFF3) [9]. The pathogenic roles of FCGBP in malignancy were firstly reported for its implication in ulcerative colitis [10], a condition that predisposes to colorectal cancer development [11]. Subsequent investigations have demonstrated the potential importance of *FCGBP* in several cancers [11,12,13,14,15,16,17]. *FCGBP* alternative splicing and *FCGBP* mutations may implicate the pathogenesis of lung cancer [12] and hepato-cholangiocarcinoma [13], respectively. Its protein is abnormally expressed in many malignancies, suggesting the potential role in tumorigenesis. In tissues from prostate adenocarcinoma in human and transgenic mice, FCGBP levels were significantly downregulated [14]. The *FCGBP* mRNA levels were significantly and differentially expressed in normal thyroid tissues, follicular- and papillary-thyroid carcinomas, with expression levels higher in adenoma and lower in carcinoma compared to that in normal tissues [15]. Moreover, the FCGBP protein served as an independent prognostic factor in gallbladder adenocarcinoma [16] and metastatic colorectal cancer [17]. The *FCGBP* level was one of the top differentially expressed transcripts between chemosensitiv and chemoresistant tissues [18]. Accordingly, advanced ovarian serous adenocarcinomas applied *FCGBP* mRNA levels to predict therapy responses [18].

In HNSC-derived cell lines, *FCGBP* levels increased by overexpressing E6 and decreased after treatment with transforming growth factor-beta (TGFβ) [19]. Similar to the conditions of the gallbladder- [16] and colorectal- [17] cancers, *FCGBP* may participate in regulating HNSC metastasis through epithelial-mesenchymal transition (EMT) [19]. Despite these findings, whether the *FCGBP* level affects immune infiltration or contributes to survival in HNSCs remains unclear. Exploring the roles of FCGBP is critical due to the tumor microenvironments’ impact on carcinogenesis and tumor progression [20]. Another concern is that immunotherapy survival advantages in an HNSC subset may be attributable to factors including immune cells, immune mediators, and related pathways [21]. Consequently, we aimed to investigate the *FCGBP* genetic characterizations on the prognostic significance, the rationale of the impacts of molecular-molecular interaction, and immune features on clinical outcomes of HNSC through bioinformatics analysis.

## 2. Materials and Methods

### 2.1. Expression Feature Analysis 

We explored the mRNA levels of *FCGBP* between cancerous and noncancerous tissues using the GEPIA2 platform [22] and evaluated the effects of *TP53* mutation and HPV infection statuses on differential *FCGBP* levels by TIMER2.0 [23] and UALCAN [24]. The correlations between diverse clinical factors and *FCGBP* levels on The Cancer Genome Atlas (TCGA) dataset were calculated by IBM SPSS Statistics for Windows, version 21 (IBM, Armonk, NY, USA). The abbreviations of different cancer types were based on TCGA Study Abbreviations (TCGA Study Abbreviations|NCI Genomic Data Commons (https://gdc.cancer.gov/, accessed on 22 March 2022). The mass spectrometry proteomic profiles generated by the Clinical Proteomic Tumor Analysis Consortium (CPTAC) were the source for protein expression analyses [25]. UALCAN database examined the correlations between the FCGBP protein level and various clinical factors. Gene/transcript expression levels are presented as log2 [transcripts per million (TPM)] or log2 (TPM + 1) in GEPIA2; log2 RNA-Seq by Expectation-Maximization (RSEM) in TIMER2.0. A *p* < 0.05 is considered as statically significant. 

### 2.2. Survival Prognosis Analysis

The prognosis analysis of the *FCGBP* level was conducted in GEPIA2 to generate the survival map and Kaplan-Meier curve for the overall- and disease-specific survival related to the *FCGBP* level from the TCGA dataset. The expression level in the top 50% (median) was considered the high-*FCGBP* group. To further test whether the prognostic significance of the *FCGBP* level was independent of other variables of HNSC, we performed univariate and multivariate Cox regressions analyses. The log-rank test compared the survival rates, with the significance threshold set at *p* < 0.05. Moreover, TIMER 2.0 generated the survival plots stratified by HPV infection.

### 2.3. Gene Alteration Analysis

We used the cBioPortal platform [26] to probe the association between different types of mutations and the *FCGBP* mRNA levels. After identifying the most frequently altered genes with *FCGBP* alteration, we estimated the genetic co-occurrence of these core genes with *FCGBP* alterations. Next, we verified the associations of the *FCGBP* level with the core genes using the “Gene_Corr module” of TIMER2.0. After submission, the correlation curve(s) was generated automatically. 

### 2.4. Molecular Interaction Analysis 

We used the STRING database [27] to explore the protein-protein interaction of the FCGBP. We identified significant proteins by intersecting with the *FCGBP* associated co-expressed genes and then tested their correlation at the mRNA level. GeneMania [28] further validated the association between FCGBP and significant proteins. GeneMania provided information for the co-expression, co-localization, gene functions, and related pathways of appealing genes in addition to the gene-gene interaction. 

### 2.5. Gene Set Enrichment Analysis

We used the LinkedOmics database [29] to download the mRNA dataset of HNSC patients and obtained a total of 517 cases containing whole *FCGBP* level and clinical features. After identifying 20,163 genes from 520 microarrays, a heat map presented the top 50 positively correlated transcripts to the *FCGBP* level. The co-expressed genes associated with the *FCGBP* level were selected for enrichment analysis to reflect the roles of FCGBP. We included a post-processing step targeting Gene Ontology (GO) to identify the most representative significant gene sets for visualization through redundancy reduction and affinity propagation cover. A *p* < 0.05 determined significantly enriched terms. Gene Set Enrichment Analysis (GSEA) also highlighted the Kyoto Encyclopedia of Genes and Genomes (KEGG) pathways through analyzing a minimum number of genes (size) of 3 and a simulation of 500 within the HNSC dataset. 

### 2.6. Immune Infiltration 

First, we analyzed the associations of the *FCGBP* level with the immunomodulatory transcripts in TISIDB [30], and identified the immune cells with significant infiltrate estimation values correlated to the *FCGBP* level. The identified immune cells were next validated in TIMER2.0 using the ‘Immune_Gene module’. We selected the immune cells that correlated significantly in the same directions using different algorithms, including TIMER, EPIC, QUANTISEQ, XCELL, MCP-COUNTER, CIBERSORT, and CIBERSORT-ABS. The associations of respective markers of the significant tumor-infiltrating immune cells with the *FCGBP* level were further explored using the TIMER2.0 correlation module. Finally, we verified the correlations of the *FCGBP* level and immune-related molecules and immune checkpoints by the TISIDB database and TIMER2.0, respectively. The Spearman correlation test analyzed the associations, and a *p*-value less than 0.05 indicates statistical significance.

## 3. Results

### 3.1. Downregulation of the FCGBP Level in Head and Neck Squamous Cell Carcinoma

To investigate the potential roles of the *FCGBP* gene in HNSC, we utilized TCGA dataset in GEPIA2, UALCAN, and TIMER2.0 to evaluate the *FCGBP* mRNA levels in tumoral and normal tissues. The *FCBGP* mRNA level from tumors was significantly lower than normal tissues (Figure 1A). However, we did not find a significant difference between HPV-related tumors (*n* = 41) and normal tissues (*n* = 44) and the *FCGBP* mRNA level (*p* = 5.2 × 10^−1^, Figure 1B). Table 1 demonstrates the correlations between the *FCGBP* level and critical clinical factors. By using the median value of *FCGBP* mRNA level to divide 442 HNSC patients into high- (*n* = 209) and low- (*n* = 233) *FCGBP* groups, we found a significant decrease in *FCGBP* level from American Joint Committee on Cancer (AJCC) T1/T2 to T3/T4 (*p* < 1 × 10^−3^) and from stage I to stage IV (*p* = 1.5 × 10^−2^) in patients with HNSC. Notably, we found a significant difference in *FCGBP* level between *TP53* mutation (*n* = 331) and wild type (*n* = 168). Furthermore, the Wilcoxon rank-sum test showed a lower *FCGBP* level associated with a higher *TP53* mutation rate (*p* = 1.8 × 10^−2^, Figure 1C). UALCAN analysis on Clinical Proteomic Tumor Analysis Consortium (CTPAC) also pointed out that the FCGBP protein levels in tumors (*n* = 108) were lower than those in normal tissues (*n* = 71) (*p* = 2.2 × 10^−8^, Figure 1D). These findings suggested the downregulation of the *FCBGP* in HNSC tissues at both transcriptional and translational levels.

### 3.2. High FCGBP Level as an Independent Prognostic Factor for Favorable Survival in HNSC

Figure 2A shows the heat map of survivals on various cancers in the pan-TCGA dataset. For HNSC, the Kaplan-Meier curves demonstrated that patients with high *FCGBP* levels have better overall survival (OS) (hazard ratio [HR) = 0.51, *p* = 9.8 × 10^−7^) and disease-free survival (DFS) (HR = 0.47, *p* = 1.1 × 10^−5^) compared to those with low *FCBGP* levels (Figure 2B,C). The univariate Cox model identified the high *FCGBP* level (*p* = 2 × 10^−3^) as a favorable prognostic factor, and high AJCC N (*p* = 2.4 × 10^−2^) classification negatively affected OS (Figure 2D). Multivariate analysis revealed that *FCGBP* level remained a significant prognostic factor, and the results are consistent in both HPV-related (*n* = 98, HR = 0.49, *p* = 1.1 × 10^−2^) and HPV-unrelated HNSC (*n* = 422, HR = 0.75, *p* = 1.4 × 10^−3^) (Figure 2E). Collectively, the above findings suggested that the *FCGBP* level may serve as a positive and independent prognostic factor in patients with HNSC.

### 3.3. The FCGBP mRNA Level Positively Correlates to TP53 WT in HNSC Patients

We analyzed genomic alteration of *FCGBP* in HNSC/TCGA, showing that the alteration rate of the *FCGBP* gene is 6%, with a mutation rate and amplification rate of 4.4% and 0.96%, respectively (Figure 3A). The *FCGBP* gene amplification was negatively associated with its mRNA level in HNSC patients (Figure 3B). Further analysis on the most frequently altered genes, which alternated with the *FCGBP* gene variations, identified *TP53, CDKN2A, titin* (*TTN*)*, FAT atypical cadherin 1* (*FAT1*)*,* and *sucrose-isomaltase* (*SI*), ranked from high to low in frequency (Figure 3C). Among these genes, *FCGBP* significantly co-occurred with *TP53* (*q* = 3.7 × 10^−2^)*, SI* (*q* = 1.9E-02)*,* and *FAT1* (*q* = 1.2 × 10^−2^) (Appendix A). Additional examination on the correlation of the *FCGBP* mRNA level and co-occurred genes with TIMER2.0 revealed that *FCGBP* significantly correlated with *TP53* mRNA levels in HNSC patients (*r* = 0.273, *p* = 6.9 × 10^−10^) (Figure 3D). Moreover, the *FCGBP* mRNA level was also positively associated with the levels of essential target transcripts of TP53, including *BCL2 binding component 3* (*BBC3*) (*r* = 0.169, *p* = 1.64 × 10^−^^4^)*, MDM2 proto-oncogene* (*MDM2*) (*r* = 0.207, *p* = 3.86 × 10^−^^6^)*, phorbol-12-myristate-13-acetate-induced protein 1* (*PMAIP1*) (*r* = 0.128, *p* = 4.39 × 10^−3^)*, RNA binding motif protein 10* (*RBM10*) (*r* = 0.226, *p* = 4.17 × 10^−^^7^), *zinc finger matrin-type 3* (*ZMAT3*) (*r* = 0.292, *p* = 3.68 × 10^−11^), and *mutL homolog 1* (*MLH1*) (*r* = 0.151, *p* = 7.94 × 10^−4^) (Appendix A). These findings suggested that a dysregulation of the *FCGBP-TP53* axis potentially participates in the pathogenesis of HNSC.

### 3.4. The FCGBP Correlated with Several Proteins in HNSC Patients

To further explore the interaction between FCGBP and potential proteins in HNSC, we used the STRING database and identified10 proteins that may interact with FCGBP (Figure 4A). Intersecting the corresponding genes of the 10 co-expressed proteins with the FCGBP-associated co-expressed genes identified by LinkedOmics, four [TFF3 (*p* = 8.27 × 10^−17^), N-acetylated alpha-linked acidic dipeptidase like 1 (NAALADL1) (*p* = 6.29 × 10^−8^), zymogen granule protein 16 (G16) (*p* = 4.72 × 10^−3^), and complement C1q C chain (C1QC) (*p* = 5.93 × 10^−3^)] were found with statistical significance (Appendix A). We next validated these proteins at the mRNA level by TIMER2.0 and found that all of these were significantly correlated to the *FCGBP* level (*p* < 5 × 10^−3^, Appendix A). 

Notably, *TFF3* ranks at the top one with a significant correlation (*r* = 0.357, *p* = 3.02 × 10^−16^) (Figure 4B). TFF3 closely interact with epidermal growth factor receptor (EGFR, combined score = 0.983) (Appendix A) through STRING analysis (Appendix A). In addition, *TFF3* downregulation in HNSC tissues, and there is a trend that high-*TFF3* is associated with better OS (Appendix A). These findings suggest that FCGBP and its co-expression with TFF3 may collectively contribute to a better survival in HNSC. GeneMania further demonstrated that *FCGBP* co-expressed and co-localized with *TFF3* (Figure 4C). Functional analysis of these differentially expressed transcripts revealed that FCGBP may relate to epithelial structure maintenance, protein O-linked glycosylation, and innate immune response (FDR < 0.05, Figure 4C). 

### 3.5. Functional Enrichment Analysis of FCGBP and Co-Expressed Genes in HNSC

To explore the potential functions and pathways of FCGBP in HNSC, we identified a total of 10,116 significantly correlated genes by the LinkedOmics database shown in volcano plot (*p* < 0.01 and FDR < 0.01, Figure 5A). Figure 5B shows the top 50 genes which were positively correlated to the *FCGBP* level. Further functional enrichment of biologic process (Figure 5C) of the GO analysis revealed that the GO function of the co-expressed genes is shown as enriched bar diagrams. The most significant enriched GO term at the biologic process level was ‘alpha-beta T cell differentiation (enrichment ratio = 2.29)’ (Figure 5C). Other GO terms include ‘immune response-activating signal transduction (enrichment ratio = 1.66)’, ‘regulation of lymphocyte activation (enrichment ratio = 1.62)’, and ‘regulation of immune response (enrichment ratio = 1.54)’ (*p* < 0.05 and FDR < 0.05, Figure 5C). Further GSEA on KEGG pathways showed that the enrichment terms of these co-expressed genes are predominantly involved in immune and inflammatory responses, including ‘Th17 cell differentiation’, ‘T cell receptor signaling pathways’, ‘cytokine-cytokine receptor interaction’, and ‘chemokine signaling pathway’, indicating that the *FCGBP* may be an immune-related factor in HNSC (FDR < 0.05, Figure 5D). 

### 3.6. The FCGBP Level Correlates to Immunomodulators and Immune-Related Molecules, and the Abundance of Infiltration of Immune Cells 

We evaluated the association of the *FCGBP* level and immunomodulators on the TISIDB database. The *FCGBP* levels positively correlated to immunostimulators across various tumors (Figure 6A). In 522 HNSC, the *FCGBP* level positively correlated to most stimulators. The top four stimulators are *CD27 molecule* (*CD27*) (*r* = 0.365, *p* = 5.44 × 10^−19^), *TNF receptor superfamily member 13B* (*TNFRSF13B*) (*r* = 0.351, *p* = 1.4 × 10^−16^), *inducible T cell costimulatory ligand* (*ICOSLG*) (*r* = 0.335, *p* = 4.78 × 10^−15^), and *TNF receptor superfamily member 17* (*TNFRSF17*) (*r* = 0.335, *p* = 5.01 × 10^−15^) (Figure 6B). These findings suggested that low *FCGBP* levels may relate to immune tolerance in HNSC since high *FCGBP* accompanied with better immune responses. 

We next performed a comprehensive screening on the infiltration of immune cells using the TISIDB database (Figure 6C). TIMER2.0 identified the meaningful infiltrated immune cells in the same directions with different algorithms. The *FCGBP* level correlated with infiltration rates of effector memory CD8+ T cells (Tem_CD8+, *r* = 0.274, *p* = 2.2 × 10^−10^), Type 17 helper cells (Th17, *r* = 0.328, *p* = 1.82 × 10^−14^), T follicular helper cells (Tfh, *r* = 0.164, *p* = 1.66 × 10^−4^), activated B cells (Act_B, *r* = 0.382, *p* < 2.2 × 10^−16^), and mast cells (Mast, *r* = 0.172, *p* = 7.81 × 10^−5^) (Figure 6D). To additionally strengthen the role of *FCGBP* in identified immune cells, we explored the association between the *FCGBP* level and the respective immune markers. Most markers were strongly associated with the *FCGBP* level in HNSC (Table 2), suggesting that *FCGBP* was strongly associated with tumor-infiltrating immune cells in HNSC. Moreover, consistent survival trends for the *FCGBP* level and involving immune cells existed. The survivals of patients with low infiltration immune cells were worse than those with high infiltration (Appendix A).

Further TISIDB analysis on the correlations between the *FCGBP* level, chemokines, and receptors showed that *FCGBP* might regulate various immune molecules (Figure 6E,F). The top two molecules and receptors with high correlations were *C-C motif chemokine ligand 22* (*CCL22*) and *CCL19*, and *C-C motif chemokine receptor 6* (*CCR6*) and *C-X3-C motif chemokine receptor 1* (*CX3CR1*), respectively. These findings further suggest that *FCGBP* and its co-expressed genes may participate in the immune response of the HNSC tumor microenvironment. In addition, we verified the relationship between *FCGBP* and immune checkpoints, which revealed that the *FCGBP* level positively associated with those of *CTLA4* (*r* = 0.295, *p* = 2.56 × 10^−11^), *HAVCR2* (*r* = 0.225, *p* = 4.41 × 10^−7^), *LAG3* (*r* = 0.162, *p* = 3.1 × 10^−4^), *PDCD1* (*r* = 0.303, *p* = 6.95 × 10^−12^) (Table 2). The differential expression levels of various immune checkpoints with *FCGBP* expression may suggest the use of immunotherapy. 

## 4. Discussion

This study found that the FCGBP can be used as an independent prognostic factor in head and neck squamous cell carcinoma, regardless of the HPV infection status. One of the potential mechanisms underlying the association of low-FCGBP with poor prognosis was downregulated FCGBP correlated to a higher TP53 mutation rate. Moreover, FCGBP correlates to TFF3, a prognostic biomarker in various cancers, at transcriptional and translational levels. An additional reason for the predictive value of FCGBP is its ability to regulate the immune responses of the HNSC tumor microenvironment, since FCGBP associated co-expressed genes were predominantly enriched in immune responses. Further, FCGBP mRNA level was significantly associated with those of various immune stimulators and immune molecules and regulated the infiltration rates of immune cells. 

The potential roles of *FCGBP* in cancer initiation, progression, and prognoses have been proposed in certain malignancies. *FCGBP* represents the most under-expressed gene in the TGF-β-induced gallbladder carcinoma-derived cells compared to a normal cell line, suggesting its role in TGFβ-induced EMT to drive metastatic behaviors [16]. However, high FCGBP protein levels in human specimens correlated to increased incidence of metastasis and better OS [16]. Moreover, the FCGBP protein levels in colorectal cancer, were lower in metastatic tissues than in the paired primary tumors, and validated as an independent prognostic factor for metastatic colon cancer [17]. In HNSC, immunohistochemistry demonstrated that the FCGBP protein levels were lower in cancer tissues than noncancerous tissues retrieved from the surgical margin [19]. Similarly, the *FCGBP* level in cancer tissues is higher in HPV-related than HPV-unrelated HNSC, and these findings are consistent with the prediction by the CTPAC analysis. Further functional analysis validated the effects of *FCGBP* on HNSC behaviors, whereas its overexpression in FaDu cells and Cal-27 cells decreased proliferation and inhibited EMT. Moreover, both mRNA and protein levels of *TGF**β* were negatively correlated to *FCGBP*, and further evidence indicates that *FCGBP* level decreased by *TGFβ* treatment and increased by E6 overexpression, suggesting that upregulated *FCBGP* may contribute to suppression of HNSC [19]. 

*TP53* may play a role in the pathogenesis of *FCGBP* in HNSC; our results showed that *TP53* was the most frequently altered gene with *FCGBP* alteration, and the tendency for the co-occurred alterations of *TP53* and *FCGBP* is significant. Further evidence strengthened the correlation between *FCGBP* and *TP53* was a significant correlation between *FCGBP* and TP53 targeted genes, including *BBC3, MDM2, PMAIP1, RBM10, ZMAT3* and *MLH1*. TP53 primarily acts as a tumor suppressor in head and neck cancer. Recent studies have pointed out that, in addition to controlling various cellular processes (e.g., apoptosis, cell cycle, senescence) in response to cellular stress to antagonized malignant progression [31], the TP53-mediated tumor suppression also involved activation of ferroptosis [32], remodeling cancer metabolism [33], inhibition of cellular self-renewal [34], ensuring genomic integrity [35], and maintenance a tumor-suppressive immune response [36]. TP53 may also modulate a target gene network to suppress tumorigenesis [35,37]. Through in vivo shRNA and CRISPR/Cas9 screens, the tumor suppressors *ZMAT3* [35,37] and *MLH1* [35] were identified as predominant TP53 target effectors to the TP53-mediated tumor suppression program. In our correlation analysis, the *FCGBP* level positively associated with *ZMAT3* and *MLH1* irrespective of the HPV status, and these findings may partially explain the inverse relationships of *FCGBP* level with AJCC T classification and AJCC stage from the perspective of the role of TP53 in HNSC. 

Current bioinformatics analysis demonstrated an inverse relationship between *TP53* mutation, the most frequently mutated gene (69.3%) in HNSC in the Pan-Cancer Atlas (TCGA), and the *FCGBP* level. The majority of *TP53* mutations are missense mutations, and the most common *TP53* mutations occur mainly in the DNA-binding domain to affect DNA binding residues or cause conformational change to prevent DNA binding [38]. The consequence includes the loss of the transcriptional function of wild-type (WT) TP53, dominant-negative activity that mutant TP53 suppresses the ability of WT TP53, and/or gain-of-function that mutant TP53 acquire oncogenic activities independently of WT TP53 [38]. Despite *TP53* mutation not necessarily causing attenuation of TP53 activity, a genomic investigation of the TCGA consisting of 279 HNSC patients identified the most significant mutations leading to TP53 inactivation were *TP53* (84%) and *CDKN2A* (57%) in HPV-unrelated HNSC [39]. Of interest, the study identified a subgroup of oral cavity cancer patients with favorable outcomes characterized by reduced copy number alteration in conjugation with inactivating *CASP8* mutations and WT *TP53* [39]. Further computational analysis of 415 HNSC patients in TCGA otherwise demonstrated *TP53* mutation is an independent prognostic factor for reduced overall survival, although the impact is affected by the types of mutations and the localization of the mutation within *TP53* [40]. More specific analyses on the transactivation activity of mutant TP53 showing the nonfunctional *TP53* mutation is indicative of reduced response to cisplatin and fluorouracil [41]. Accordingly, these findings may suggest that *FCGBP* downregulation in HNSC predicted worse survivals to a certain extent, especially in the HPV-unrelated HNSC since *TP53* is rarely altered in HPV-related HNSC [39].

Another possible mechanism for low *FCGBP* as an inferior prognostic factor in HNSC is TFF3 might interact with FCGBP. In the genetic network analysis, *FCGBP* and *TFF3* are co-expressed and co-localized, revealing their associations at transcriptional and translational levels. TFF3 is a secretory lectin and represents the predominant TFF peptide of human saliva and esophageal secretions [42,43]. Physiologically, TFF3 may bind FCGBP to form a heterodimer to protect the mucosa by reinforcing innate immunity [9,44]. Interestingly, *TFF3* has multifaceted effects from a pre-neoplastic lesion to invasive cancer, and the roles in several cancer subtypes have been extensively discussed [45,46,47]. In colorectal cancer, TFF3 promotes proliferation, invasion, and migration by enhancing CD147–CD44 interaction to activate transcription 3 (STAT3) and prostaglandin G/H synthase 2 (PTGS2) expression [46]. In prostate cancer, *TFF3* silencing induced mitochondria-mediated apoptosis, thus suppressing tumor growth and migration [47]. The roles of *TFF3* in promoting tumor progression were also well established in breast, cervical, hepatocellular, and gastric cancers and glioblastoma. However, *TFF3* contrarily acts as a tumor suppressor in retinoblastoma [45]. Since protein–protein interaction is essential in the molecular mechanisms underlying carcinogenesis and cancer progression and aggressiveness, aberrant FCGBP expression may regulate cancer behaviors by functional interactome linking with TFF3. Our results may potentially rationalize the assumption because *TFF3* level, such as *FCGBP* expression, was reduced in HNSC tissue, and survival decreased in patients with HNSC of downregulated *TFF3*. Another supporting finding is that *FCGBP* in thyroid cancer can functionally synergize with *TFF3* through genetic co-expression, and thus contributing to poor survival in thyroid cancer patients with low *TFF3* level [48].

In addition, *FCGBP* may participate in the immune response in the tumor microenvironment of HNSC. Our GO analysis results demonstrated that the *FCGBP*-associated co-expressed genes were primarily enriched in immune and inflammatory responses, including ‘regulation of lymphocyte activation’, ‘regulation of immune response’, and ‘immune response-activating signal transduction’ for GO at the biological process level. Our GSEA results of the KEGG pathway also revealed that the enriched terms are predominantly related to the immune process, including ‘T cell receptor signaling pathways’, ‘cytokine–cytokine receptor interaction’, and ‘chemokine signaling pathways’. Altogether, these findings imply that *FCGBP* levels may be related to the immune interaction with HNSC. *FCGBP* may be involved in the tumor microenvironment of HNSC to regulate cancer development because cytokines and chemokines are critical noncellular components of the tumor microenvironment [49].

Further exploration of the association of *FCGBP* expression with diverse immune molecules and immune-stimulator genes may support the assumption that most of them are significantly correlated to *FCGBP* level. Thus, we analyzed the composition of tumor-infiltrating immune cells correlating to *FCGBP* levels in HNSC samples. The *FCGBP* level was positively associated with infiltration rates of CD8+ T-cell, T17 lymphocyte, B-cell, mast cell, and follicular helper T-cell. The correlation was strengthened with the strong associations of the FCGBP level with those of most of these cells’ type-specific markers irrespective of the HPV infection status. Previous investigations have pointed out that the dense infiltration of the CD8+ T cells within tumor microenvironment confers a favorable prognosis for its antitumor property [50,51], we hypothesized that a low FCGBP level in the tumor tissue might shape the tumor microenvironment in an immune-suppressed state. There is a consistent survival trend between the *FCGBP* level and immune cell abundance, i.e., the survival is worse in low-abundant immune cells. Consequently, the predictive value of *FCGBP*, at least, might result from better immune control from the perspective of immune infiltration. 

This study has some limitations. First, the study used heterogeneous datasets in different databases; therefore, the analysis by the other platforms may produce inconsistent results. Except for GeneMania and STRING, all other studies used available data from TCGA with different platforms. Because the required raw data for each analysis may be unavailable in the cohort, the total number of HNSC recruited differs by different analyses. Nonetheless, despite the slight difference in recruited samples in each analysis, the overall trend for *FCGBP* was present. For example, *FCGBP* was an independent prognostic factor for both HPV-related and HPV-unrelated HNSCs since the HPV status may cause a distinct difference in cancer behaviors of HNSC. Second, although other datasets have validated the findings of TCGA; the results should be extrapolated cautiously to non-primary ethnicities. Third, we explored the role of *FCGBP* in HNSC from a survival perspective, the effects of chemotherapy and radiation therapy cannot be ignored. Although we can forecast drug sensitivity from bioinformatics prediction, the complexity of treatment, for example, treatment intensity or treatment sequence may make us overestimate or underestimate the role of FCGBP. Future experiments focusing on validating the present predictions are warranted to improve the value of FCGBP application in HNSC.

## 5. Conclusions

Taken together, *FCGBP* level was significantly reduced in HNSC tissues, and a high-*FCBGP* level indicated favorable prognoses. Further analyses demonstrated that *FCGBP* alteration significantly co-occurred with that of *TP53*, and *FCGBP* levels were negatively associated with the *TP53* mutation rate. Additionally, FCGBP might interact with TFF3, an essential protein in the pathogenesis of several cancers. With the finding that *FCGBP* level significantly correlates with infiltration rates of various immune cells and expression levels of immune molecules to possibly regulate the immune response of the tumor microenvironment, *FCGBP* may be a potential prognostic biomarker in head and neck squamous cell carcinoma. However, further studies are needed to clarify these.

## Figures and Tables

**Figure 1 diagnostics-12-01178-f001:**
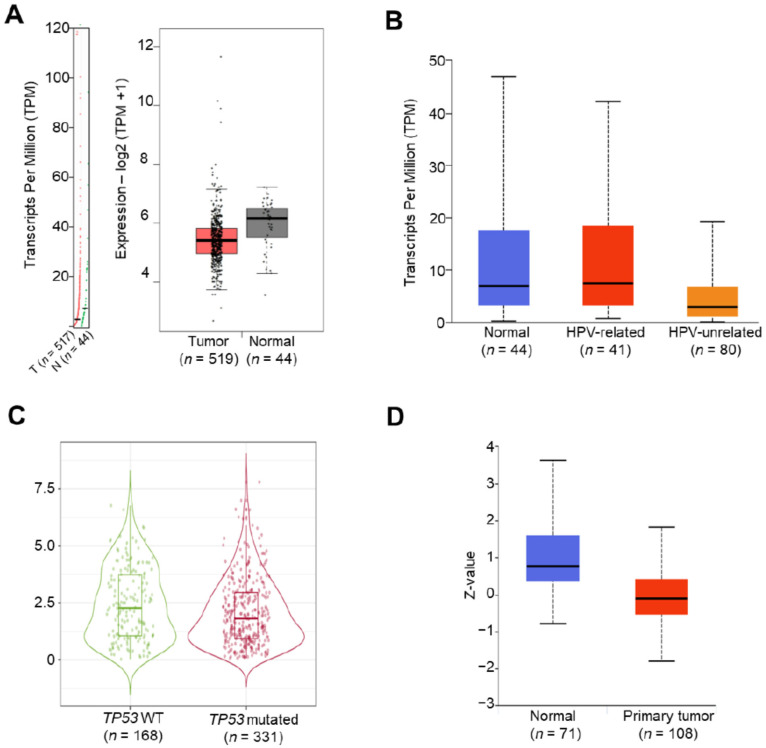
The *FCGBP* levels in head and neck squamous cell carcinoma (HNSC) patients. (**A**) Gene Expression Profiling Interactive Analysis 2 (GEPIA2) illustrated the differential expression levels of the *FCGBP* transcript in tumor and adjacent normal tissues based on HNSC dataset. (**B**) UALCAN showed the *FCGBP* levels in normal tissues, HPV-related (*n* = 41) and HPV-unrelated (*n =* 80) HNSC. (**C**) TIMER 2.0 identified the *FCGBP* levels in HNSC with *TP53* wild-type (WT) and mutated, respectively. (**D**) The protein level of FCGBP in cancer (*n* = 108) and non-cancerous tissues (*n* = 71) were shown, based on Clinical Proteomic Tumor Analysis Consortium (CPTAC).

**Figure 2 diagnostics-12-01178-f002:**
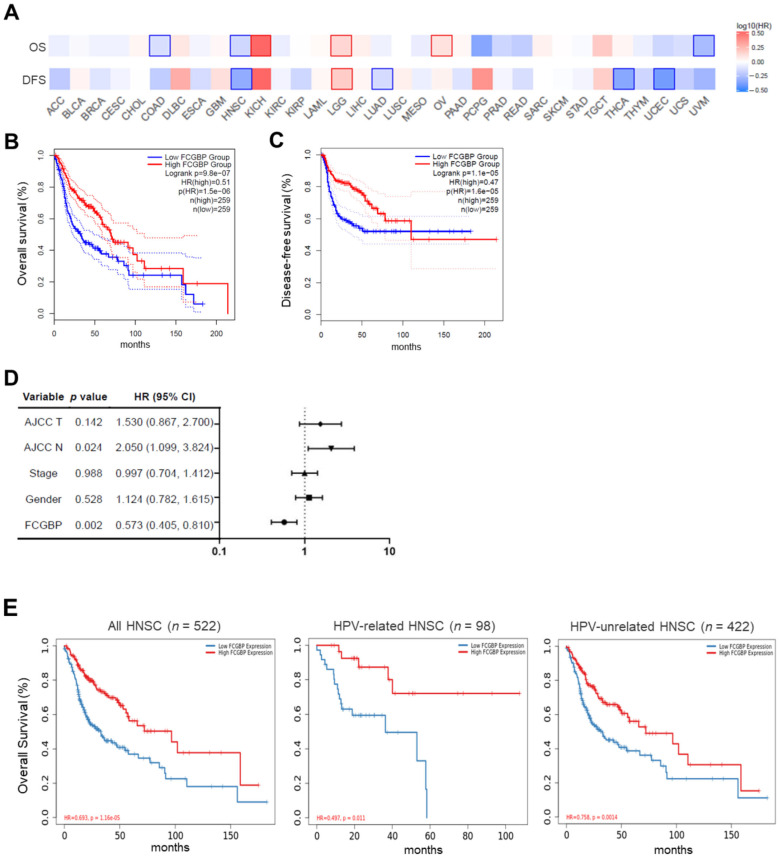
Survival analyses of the *FCGBP* level in patients with HNSC. (**A**) Gene Expression Profiling Interactive Analysis 2 (GEPIA2) illustrated the heat maps of the overall survival (OS) and disease-free survival (DFS) across various cancer types. (**B**,**C**) Kaplan–Meier analysis showed OS and DFS of HNSC patients with high *FCGBP* and low *FCGBP* levels based on the GEPIA2. (**D**) Univariate Cox analysis reveals the hazard ratios (HR) of different variables. (**E**) Using multivariate analysis, TIMER 2.0 evaluates the OS for all HNSC, HPV-related, and HPV-unrelated HNSC.

**Figure 3 diagnostics-12-01178-f003:**
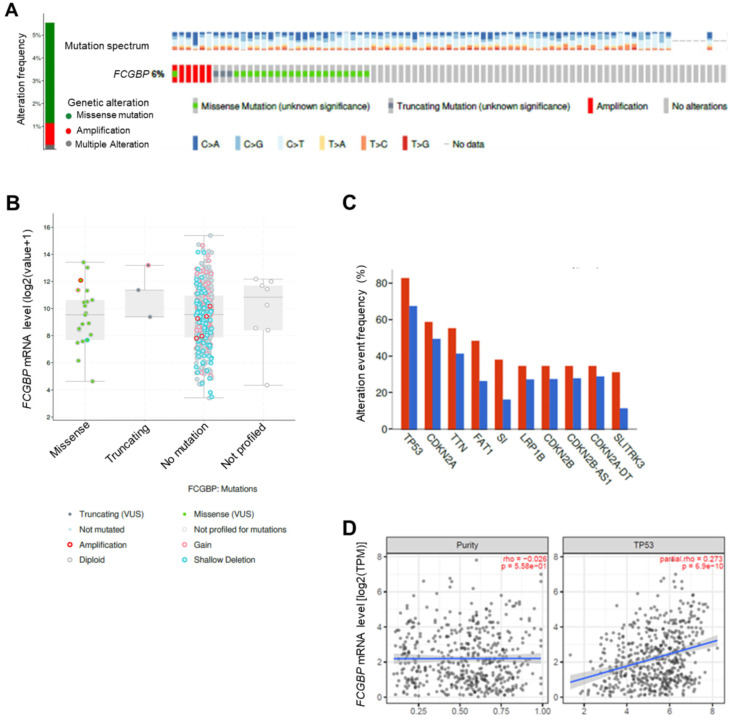
Genetic alterations and correlation analysis on the *FCGBP* in HNSC patients. (**A**) cBioPortal showed the *FCGBP* genetic alterations and mutation frequencies. Different colors indicate dissimilar types of genetic alterations. (**B**) The relationships between the *FCGBP* variants and their corresponding mRNA levels (RNA-Seq by Expectation-Maximization: RSEM), batch normalized from Illumina Hiseq_RANSeqV2. (**C**) The most frequently altered genes with *FCGBP* alteration in HNSC (*n* = 523). (**D**) TIMER2.0 analyzed the correlations between *FCGBP* mRNA level and tumor purity (**left**) and *TP53* mRNA level (**right**).

**Figure 4 diagnostics-12-01178-f004:**
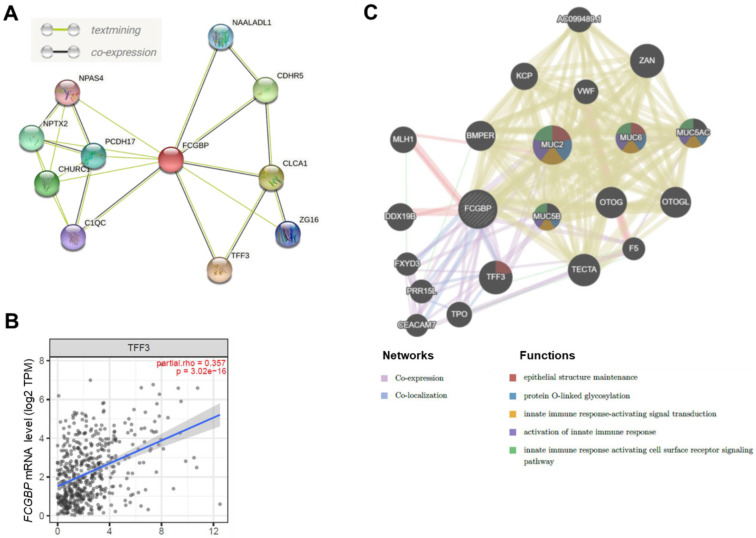
The FCGBP interacted gene/protein networks. (**A**) The STRING database showed the FCGBP interacted gene/protein network based on text mining and co-expression. (**B**) TIMER2.0 identified a potential interaction between FCGBP and TTF3 based on co-expression in HNSC patients (*n* = 522; partial *r* = 0.357, correlation adjusted by purity; *p* < 0.001). (**C**) The GeneMania database showed the gene-gene interaction network. The association between *FCGBP* and *TFF3* based on co-expression and co-localization highlighted the genes with function enriched in epithelial structure maintenance.

**Figure 5 diagnostics-12-01178-f005:**
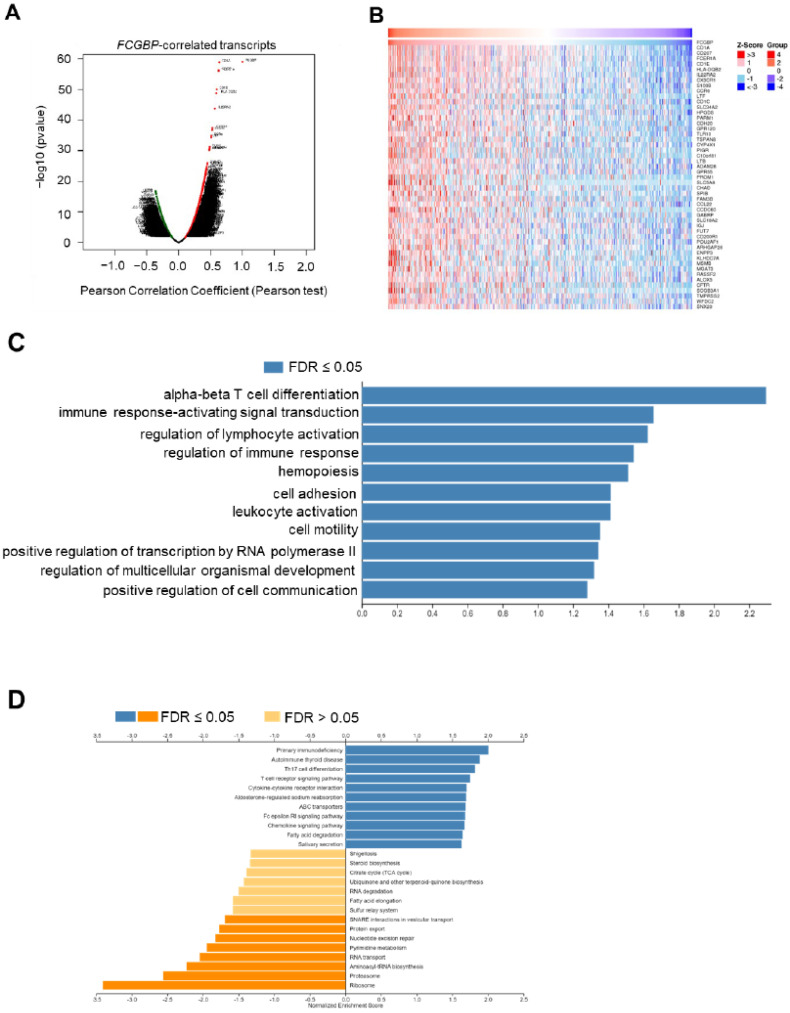
The *FCGBP*-correlated transcripts and enrichment analysis. (**A**) LinkedOmics showed the volcano map of 20,163 *FCGBP* co-expressed transcripts in 517 HNSCs by the Pearson correlation analysis (red: positive; green: negative). (**B**) The heat maps displayed the top 50 transcripts positively correlated to the *FCGBP* level. (**C**) The bar diagram demonstrated the co-expressed transcripts with significantly enriched GO (Biological Process) annotations (*p* < 0.05 and FDR < 0.05). (**D**) Gene Set Enrichment Analysis (GSEA) showed the enriched KEGG pathways (FDR < 0.05).

**Figure 6 diagnostics-12-01178-f006:**
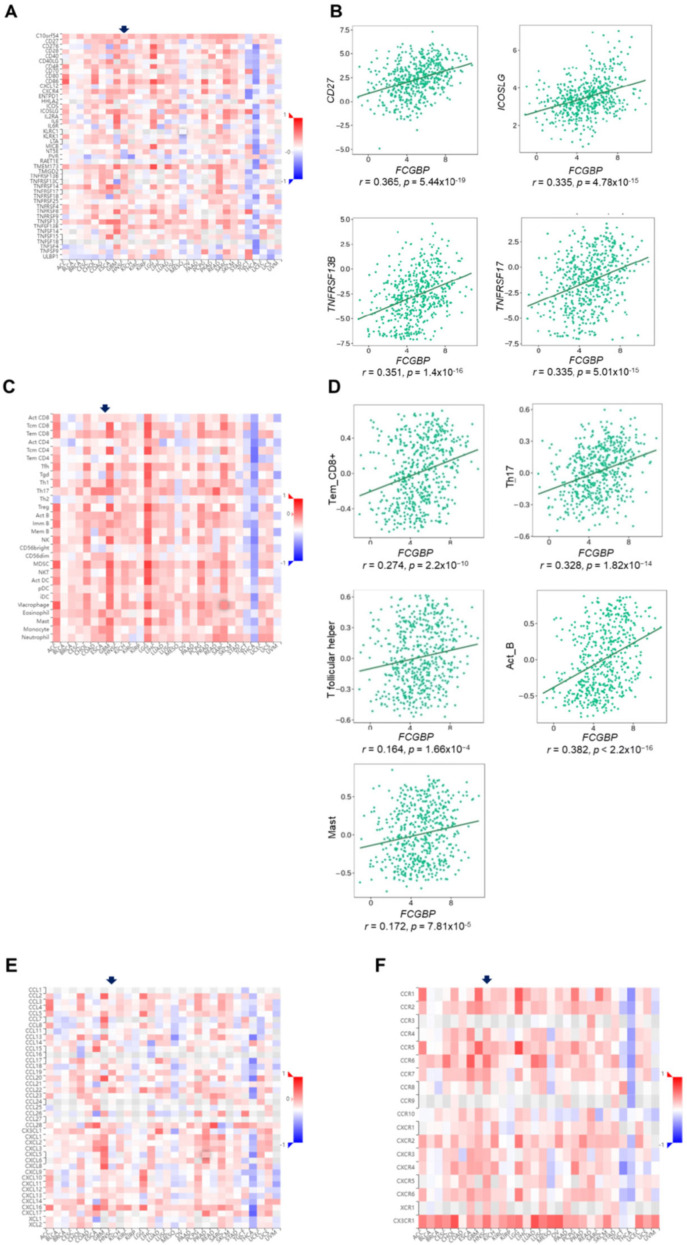
TISIDB demonstrates the correlations between HNSC and the *FCGBP* level and various immunostimulators, chemokines and tumor-infiltrating cells. (**A**) A heat map with the correlations (*r* = −1 to +1) between various immunostimulators and the HNSC (arrow) among distinct cancer types. (**B**) In HNSC patients (*n* = 522, TCGA), the correlations (*r*) between the *FCGBP* level and the *CD27*, *TNFRSF13B*, *ICOSLG*, and *TNFRSF17* levels, respectively. (**C**) Heat maps indicated the correlations (*r* = −1 to +1) between various infiltration immune cells and the HNSC (arrow) among distinct cancer types. (**D**) In HNSC 522 patients (TCGA), the correlations (*r*) between the *FCGBP* level and abundance of effector memory CD8 T cell (Tem_CD8), type 17 helper cell (Th17), T follicular helper cells (*Tfh*), activated B cells (Act_B), and mast cells (Mast). (**E**,**F**) Heap maps of chemokines, chemokine receptors and the HNSC (arrow) among distinct cancer types.

**Table 1 diagnostics-12-01178-t001:** Correlations between clinical factors and the *FCGBP* mRNA levels.

Variables	*FCBGP* mRNA Level		
	Low	High	*p* Value
	*n* = 233	*n* = 209	
Gender			1.0
Male	170	152	
Female	63	57	
AJCC ^1^ T Classification			<0.001
T1/T2	72	99	
T3/T4	161	110	
AJCC ^1^ N Classification			0.894
N0	106	97	
N1/N2/N3	127	112	
Stage			0.015
Stage I	7	20	
Stage II	33	37	
Stage III	43	38	
Stage IV	150	114	

^1^ AJCC: American Joint Committee of Cancer.

**Table 2 diagnostics-12-01178-t002:** Associations between FCGBP level and gene markers in tumor-infiltrating immune cells in HNSC by HPV infection status.

Cell Type	Marker	^1^ HNSC (*n* = 522)				^2^ HPV-Unrelated HNSC (*n* = 422)				HPV-Related HNSC (*n* = 98)			
		^3^ Purity		^4^ None		Purity		None		Purity		None	
		^5^ Correlation	*p* Value	Correlation	*p* Value	Correlation	*p* Value	Correlation	*p* Value	Correlation	*p* Value	Correlation	*p* Value
B													
	CD19	0.38	***	0.38	***	0.32	***	0.33	***	0.37	***	0.32	**
	CD79A	0.42	***	0.41	***	0.35	***	0.36	***	0.43	***	0.37	***
	CD86	0.19	***	0.20	***	0.18	***	0.19	***	0.19	7.13 × 10^−2^	0.15	1.33 × 10^−1^
	CSF1R	0.34	***	0.33	***	0.32	***	0.31	***	0.38	***	0.33	**
Plasma													
	CD38	0.11	*	0.13	**	0.08	1.12 × 10^−1^	0.09	6.82 × 10^−2^	0.23	*	0.13	**
	CXCR4	0.31	***	0.32	***	0.26	***	0.27	***	0.34	**	0.35	***
	TNFRSF17	0.37	***	0.37	***	0.31	***	0.32	***	0.39	***	0.34	***
	CD27	0.41	***	0.40	***	0.35	***	0.35	***	0.44	***	0.38	***
CD8+ T													
	CD8A	0.28	***	0.29	***	0.21	***	0.23	***	0.38	***	0.35	***
	CD8B	0.28	***	0.29	***	0.20	***	0.22	***	0.40	***	0.41	***
Follicular helper T													
	IL21	0.23	***	0.24	***	0.16	**	0.18	***	0.34	**	0.32	**
	BCL6	0.30	***	0.28	***	0.29	***	0.26	***	0.25	1.71 × 10^−2^	0.22	*
	ICOS	0.29	***	0.29	***	0.25	***	0.26	***	0.32	**	0.32	**
	CXCR5	0.42	***	0.41	***	0.38	***	0.38	***	0.37	***	0.33	***
Th17													
	STAT3	0.37	***	0.37	***	0.32	***	0.31	***	0.50	***	0.49	***
	IL-17A	0.34	***	0.34	***	0.26	***	0.28	***	0.49	***	1.74	***
	IL-21R	0.41	***	0.40	***	0.35	***	0.35	***	0.48	***	1.47	***
	IL-23R	0.31	***	0.31	***	0.27	***	0.27	***	0.38	***	0.39	***
T cell exhaustion													
	PDCD1	0.30	***	0.31	***	0.22	***	0.24	***	0.42	***	0.39	***
	CTLA4	0.30	***	0.30	***	0.23	***	0.25	***	0.40	***	0.39	***
	LAG3	0.16	***	0.17	***	0.09	6.85 × 10^−2^	0.11	*	0.29	**	0.28	**
	HAVCR2	0.23	***	0.22	***	0.18	***	0.19	***	0.29	**	0.23	*

^1^ HNSC: head and neck squamous cell carcinoma, ^2^ HPV: human papillomavirus, ^3^ Purity: correlation adjusted by tumor purity; ^4^ None: correlation without adjustment; ^5^ Correlation: R value of Spearman’s correlation. * *p* < 0.05, ** *p* < 0.01, *** *p* < 0.001.

## Data Availability

The data presented in this study are available on request from the corresponding author.

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
