# Peer review of "Multi-Omics Analyses to Identify FCGBP as a Potential Predictor in Head and Neck Squamous Cell Carcinoma"

_diagnostics, 2022, doi:10.3390/diagnostics12051178_

Round 1
Reviewer 1 Report
In this manuscript, several databases/platforms were used to evaluate FCGBP significance in head and neck cancer. There are several concerns and some of them listed below.
Some statements should be corrected; for example: “oncoprotein E7 inactivated RB1 through phosphorylation” is not accurate. Another example is “Because TP53 may affect the cancer cell cycle through the p53/TGFβ signaling pathway, TP53 mutation is usually associated with aggressive cancer behaviors and short survival time in HNSC”. First of all, TGFβ signaling pathway is not the only and not the first pathway through which p53 affects the cell cycle. And of course, this is not the only reason why p53 is mutated in cancer. I strongly suggest the authors to perform an extensive literature review about all tumors suppressors and oncogenes they mentioned in the manuscript and adjust their statements.
“Notably, we found a significant difference in FCGBP level between TP53 mutation (n = 337) and wild type (n = 172)”. The number of tumors with wtTP53 seems very high; TCGA cohort contains 353 tumors with altered TP53 and 143 tumors with wt including HPV+ cases.
Are HPV+ patients included in the survival analysis? Also, it is very difficult to understand throughout the text which cohort of patients is used and it should be clearly stated. It looks that different number of tumors is used in different figures, sometime including HPV+, sometime not.
Author Response
Dear Editor,
We herein response the reviewers’ comments, point-by-point,
Re: Reviewer 1
Comment 1
Some statements should be corrected; for example: “oncoprotein E7 inactivated RB1 through phosphorylation” is not accurate.
- Thank you for bringing this to our attention. The E7 protein may bind strongly to the cell cycle regulator retinoblastoma-associated protein (RB1) to promote proteasomal degradation of RB1 and the release of E2F family transcription factors. The liberated E2F proteins drive the cell cycle beyond the restriction point and into the S phase. The dysfunctional RB1 by E7 leads to a feedback upregulation of the CDK4/6 inhibitor p16INK4A which is encoded by tumor suppressor CDKN2A.
- To clarify this, we have revised this sentence into ‘another oncoprotein E7 eliminated RB1 through proteasomal degradation of RB1 to facilitate cell cycle progression, thereby transactivation of the CDKN2A via E2F proteins releasing from RB1/ E2F family transcription factors [4]’ (page 2, line 51 to line 54, highlighted in blue fonts)
Comment 2
Another example is “Because TP53 may affect the cancer cell cycle through the p53/TGFβ signaling pathway, TP53 mutation is usually associated with aggressive cancer behaviors and short survival time in HNSC”. First of all, TGFβ signaling pathway is not the only and not the first pathway through which p53 affects the cell cycle. And of course, this is not the only reason why p53 is mutated in cancer.
- Thank you very much for bringing this to our attention. To be more comprehensive, we have disassembled our discussion of FCGBP with TP53 level and TP53 mutation into two paragraphs.
- ‘TP53 may play a role in the pathogenesis of FCGBP in HNSC; our results showed that TP53 was the most frequently altered gene with FCGBP alteration, and the tendency for the co-occurred alterations of TP53 and FCGBP is significant. Further evidence strengthened the correlation between FCGBP and TP53 was a significant correlation between FCGBP and TP53 targeted genes, including BBC3, MDM2, PMAIP1, RBM10, ZMAT3 and MLH1. TP53 primarily acts as a tumor suppressor in head and neck cancer. Recent studies have pointed out that, in addition to controlling various cellular processes (e.g., apoptosis, cell cycle, senescence) in response to cellular stress to antagonized malignant progression [31], the TP53-mediated tumor suppression also involved activation of ferroptosis [32], remodeling cancer metabolism [33], inhibition of cellular self-renewal [34], ensuring genomic integrity [35], and maintenance a tumor-suppressive immune response [36]. TP53 may also modulate a target gene network to suppress tumorigenesis [35,37]. Through in vivo shRNA and CRISPR/Cas9 screens, the tumor suppressors ZMAT3 [35,37] and MLH1 [35] were identified as predominant TP53 target effectors to the TP53-mediated tumor suppression program. In our correlation analysis, the FCGBP level positively associated with ZMAT3 and MLH1 irrespective of the HPV status, and these findings may partially explain the inverse relationships of FCGBP level with AJCC T classification and AJCC stage from the perspective of the role of TP53 in HNSC.
- Current bioinformatics analysis demonstrated an inverse relationship between TP53mutation, the most frequently mutated gene (69.3%) in HNSC in the Pan-Cancer Atlas (TCGA), and the FCGBP. The majority of TP53 mutations are missense mutations, and the most common TP53 mutations occur mainly in the DNA-binding domain to affect DNA binding residues or cause conformational change to prevent DNA binding [38]. The consequence includes the loss of the transcriptional function of wild-type (WT) TP53, dominant-negative activity that mutant TP53 suppresses the ability of WT TP53, and/or gain-of-function that mutant TP53 acquire oncogenic activities independently of WT TP53 [38]. Despite TP53 mutation not necessarily causing attenuation of TP53 activity, a genomic investigation of the TCGA consisting of 279 HNSC patients identified the most significant mutations leading to TP53 inactivation were TP53 (84%) and CDKN2A (57%) in HPV-unrelated HNSC [39]. Of interest, the study identified a subgroup of oral cavity cancer patients with favorable outcomes characterized by reduced copy number alteration in conjugation with inactivating CASP8 mutations and WT TP53 [39]. Further computational analysis of 415 HNSC patients in TCGA otherwise demonstrated TP53 mutation is an independent prognostic factor for reduced overall survival, although the impact is affected by the types of mutations and the localization of the mutation within TP53 [40]. More specific analyses on the transactivation activity of mutant TP53 showing the nonfunctional TP53 mutation is indicative of reduced response to cisplatin and fluorouracil [41]. Accordingly, these findings may suggest that FCGBP downregulation in HNSC predicted worse survivals to a certain extent, especially in the HPV-unrelated HNSC since TP53 is rarely altered in HPV-related HNSC [39]. (page 17, lines 411 to page 18, line 447)
- References
- Vousden, K.H.; Prives, C. P53 and prognosis: new insights and further complexity. Cell 2005, 120, 7–10.
- Chu, B.; Kon, N.; Chen, D.; Li, T.; Liu, T.; Jiang, L.; Song, S.; Tavana, O.; Gu, W. ALOX12 is required for p53-mediated tumour suppression through a distinct ferroptosis pathway. Nat Cell Biol. 2019, 21, 579-591.
- Morris, J.P. 4th; Yashinskie, J.J.; Koche, R.; Chandwani, R.; Tian, S.; Chen, C.C.; Baslan, T.; Marinkovic, Z.S.; Sánchez-Rivera, F.J.; Leach, S.D.; et al. α-Ketoglutarate links p53 to cell fate during tumour suppression. Nature. 2019, 573, 595-599.
- Spike, B.T.; Wahl, G.M. p53, Stem Cells, and Reprogramming: Tumor Suppression beyond Guarding the Genome. Genes Cancer. 2011, 2, 404-419.
- Janic, A.; Valente, L.J.; Wakefield, M.J.; Di Stefano, L.; Milla, L.; Wilcox, S.; Yang, H.; Tai, L.; Vandenberg, C.J.; Kueh, A.J.; et al. DNA repair processes are critical mediators of p53-dependent tumor suppression. Nat Med. 2018, 24, 947-953.
- Blagih, J.; Zani, F.; Chakravarty, P.; Hennequart, M.; Pilley, S.; Hobor, S.; Hock, A.K.; Walton, J.B.; Morton, J.P.; Gronroos, E.; et al. Cancer-Specific Loss of p53 Leads to a Modulation of Myeloid and T Cell Responses. Cell Rep. 2020, 30, 481-496.
- Bieging-Rolett, K.T.; Kaiser, A.M.; Morgens, D.W.; Boutelle, A.M.; Seoane, J.A.; Van Nostrand, E.L.; Zhu, C.; Houlihan, S.L.; Mello, S.S.; Yee, B.A.; et al. Zmat3 Is a Key Splicing Regulator in the p53 Tumor Suppression Program. Mol Cell. 2020, 80, 452-469.
- Zhou, G.; Liu, Z.; Myers, J.N. TP53 Mutations in Head and Neck Squamous Cell Carcinoma and Their Impact on Disease Progression and Treatment Response. J Cell Biochem. 2016, 117, 2682-2692.
- Cancer Genome Atlas Network. Comprehensive genomic characterization of head and neck squamous cell carcinomas. Nature. 2015, 517, 576-582.
- Caponio, V.C.A.; Troiano, G.; Adipietro, I.; Zhurakivska, K.; Arena, C.; Mangieri, D.; Mascitti, M.; Cirillo, N.; Lo Muzio, L. Computational analysis of TP53 mutational landscape unveils key prognostic signatures and distinct pathobiological pathways in head and neck squamous cell cancer. Br J Cancer. 2020, 123, 1302-1314.
- Perrone, F.; Bossi, P.; Cortelazzi, B.; Locati, L.; Quattrone, P.; Pierotti, M.A.; Pilotti, S.; Licitra, L. TP53 mutations and pathologic complete response to neoadjuvant cisplatin and fluorouracil chemotherapy in resected oral cavity squamous cell carcinoma. J Clin Oncol. 2010, 28, 761-766.
- Additionally, we described the correlations of FCGBP level with ZMAT3 and MLH1, the target gene effectors to the TP53-mediated tumor suppression network. The relevant information was revised on page 8, line 237 to line 238; page 17, line 411, and the Supplementary Fig. S1.
Comment 3
I strongly suggest the authors to perform an extensive literature review about all tumors suppressors and oncogenes they mentioned in the manuscript and adjust their statements.
- Thank you for your penetrative suggestion. We have extensively reviewed the literature regarding the genes we have mentioned in the manuscript. CKDN2A and FAT1, the most altered genes altered with FCGBP, are tumor suppressors. On the other hand, the role of TP53 and its targeted genes have been addressed in the previous comment. However, the correlation- and p-values for the target genes of TP53 have not been adjusted by tumor purity, and we have made corrections on page 8, line 234 to line 237 and the Supplementary Fig. S1.
- As for immune-related genes, we have mentioned in the section Methodology that we use the TISIDB database to analyze the role of FCGBP in tumor-immune interaction. The web portal TISIDB integrated multiple types of data resources in oncoimmunology. After manually curating 4176 records from 2530 publications which reported 988 genes related to anti-tumor immunity, genes associated with the resistance or sensitivity of tumor cells to T cell-mediated killing and immunotherapy were identified by analyzing high-throughput screening and genomic profiling data. The association between genes and immune features, such as lymphocytes, immunomodulators, and chemokines, was pre-calculated for 30 TCGA cancer types. Therefore, we may cross-check the gene of interest about its role in tumor-immune interaction through literature mining and high-throughput data analysis. We have confirmed the accuracy of each gene mentioned in the section Result, ‘The FCGBP level correlates to immunomodulators and immune-related molecules, and the abundance of infiltration of immune cells.’
- As for the role of TFF3in cancer biology, it varies by different cancer types. For example, evidence demonstrates the critical role of TFF3 in tumor progression and metastasis, extending their actions beyond protection. To clarify this, we have emphasized the role of TFF3. ‘The roles of TFF3 in promoting tumor progression were also well established in breast, cervical, hepatocellular, and gastric cancers and glioblastoma. However, TFF3 contrarily acts as a tumor suppressor in retinoblastoma [45]. Since protein–protein…’ (page 18, line 460 to line 463)
Reference
- Jahan, R.; Shah, A.; Kisling, S.G.; Macha, M.A.; Thayer, S.; Batra, S.K.; Kaur, S.Odyssey of trefoil factors in cancer: Diagnostic and therapeutic implications. Biochim Biophys Acta Rev Cancer. 2020, 1873, 188362.
Comment 4
“Notably, we found a significant difference in FCGBP level between TP53 mutation (n = 337) and wild type (n = 172)”. The number of tumors with wtTP53 seems very high; TCGA cohort contains 353 tumors with altered TP53 and 143 tumors with wt including HPV+ cases.
- Thank you very much for your detailed review to point this out. Figure 1C describes the mutation status of TP53 and the expression level of FCGBP. The violin plot generated from the ‘Mutation Module’ of TIMER 2.0 displaced a significant difference in FCGBP level between tumors with mutant or WT TP53 in HNSC.
- According to the data offered by TIMER 2.0 (please find the attachment figure), the number for mutated TP53 is 331, and 168 for the wild-type in the correlation analysis. We have revised the typos in the sentence’ Notably, we found a significant difference in FCGBP level between TP53 mutation (n = 331) and wild type (n = 168)’ (page 4, line 179) and the revised Figure 1C in page 5.
Comment 5
Are HPV+ patients included in the survival analysis? Also, it is very difficult to understand throughout the text which cohort of patients is used and it should be clearly stated. It looks that different number of tumors is used in different figures, sometime including HPV+, sometime not.
- The analyses in the study exempted the use of the Cancer Genome Atlas (TCGA) was the establishment of gene-gene- and protein-protein-interactions explored by GeneMania and STRING database. Other platforms, including cBioPortal, GEPIA2, UALCAN, and TIMER 2.0, primarily used TCGA cohort data to perform various gene-signature-based analyses. For example, aside from estimating the tumor-immune interactions, the TIMER 2.0 platform may offer several modules that allow users to find cancer-related features of interest in TCGA. However, the available raw data required differ for different analyses; the recruited patients in each analysis are not the same. In the present study, the total number of HNSC patients in further analyses ranged from 499 to 523, except for the protein expression differences in cancer- and non-cancer tissues consisting of only 108 tumors since the analysis is generated by the Clinical Proteomic Tumor Analysis Consortium (CPTAC) on selected TCGA tumor samples.
- Despite this, it should be confident that HPV-positive patients were included in each analysis. In the association analysis of TP53 mutation and FCGBP level that included the least total number of HNSC patients (n = 499), the platform timer 2.0 pointed out that 92 HNSC patients were HPV-related (attachment figure in comment 4). Nevertheless, the reviewer’s comment is crucial since HPV-related HNSCs exhibit distinct differences from HPV-unrelated HNSCs in gene expression and mutational and immune profiles, so we might not exclude the possibility that the varying numbers may affect the analytic results.
- To clarify this, we have listed the varying patient number as our study limitation.
‘Except for GeneMania and STRING, all other studies used available data from TCGA with different platforms. Because the required raw data for each analysis may be unavailable in the cohort, the total number of HNSC recruited differs by different analyses. Nonetheless, despite the slight difference in recruited samples in each analysis, the overall trend for FCGBP was present. For example, FCGBP was an independent prognostic factor for both HPV-related and HPV- unrelated HNSCs since the HPV status may cause a distinct difference in cancer behaviors of HNSC.’(page 19, line 504 to line 509)
We deeply appreciate the valuable time from reviewers and editors. Looking forward to knowing this manuscript is acceptable by the Journal.

Reviewer 2 Report
The authors use public head and neck cancer sequencing databases to investigate the correlation of FCGBP with head and neck cancer. They identify decreasing FCGBP associated with head and neck cancer versus normal tissue, and with advanced stages. They provide an interesting story for the expression of FCGBP and its association with head and neck cancer. The use of the databases and analysis seem appropriate. The language and grammar is appropriate.
Comments:
- How did the authors delineate "high" and "low" expression levels? e.g. in Table I. Did they use specific cut-offs on these levels?
- There seem to be some errors in Table I. AJCC N Stage lists "T3/T4". I would assume this should be an N staging category. Also, Stage does not include stage IV, although there seem to be T4 tumors, which would be stage IV.
Author Response
Dear Editor,
We herein response the reviewers’ comments, point-by-point,
Re: Reviewer 2
Comment 1
How did the authors delineate "high" and "low" expression levels? e.g. in Table I. Did they use specific cut-offs on these levels?
- Thank you for bringing this to our attention. We download the raw data of the TCGA-HNSC cohort (n= 523). To demonstrate the correlation of FCGBP level with clinical variables, we exclude HNSC patients with
- unclear AJCC T- and/or AJCC N- classifications;
- metastasis at the initial presentation;
- inappropriate follow-up period (e.g., the date of the last follow-up is earlier than that of diagnosis).
After exclusion, a total of 442 patients were left for analysis. We used the median value of FCGBP mRNA level of the 442 HNSC patients to divide them into high- (n = 209) and low- (n = 233) FCGBP groups. The KM-curve for overall- and disease-specific survival related to FCGBP level by GEPIA2 also used the median value as the cut-off, and we have already mentioned in the Section Materials and Methods.
- To clarify, we added the information.
‘…factors. By using the median value of FCGBP mRNA level to divide 442 HNSC patients into high- (n = 209) and low- (n = 233) FCGBP groups, we found a significant decrease in FCGBP level from American Joint Committee on Cancer (AJCC) T1/T2 to T3/T4 (p < 1E-03) and from stage I to stage IV (p = 1.5E-02) in patients with HNSC.’ (page 4, line 174 to line 176)
Comment 2
There seem to be some errors in Table I. AJCC N Stage lists "T3/T4". I would assume this should be an N staging category. Also, Stage does not include stage IV, although there seem to be T4 tumors, which would be stage IV.
- Thank you for you to point this out. These categories are typos, and they should be N1/N2/N3. We have made corrections in the revised Table 1 in page 5.
- Additionally, because of the layout, the table is displaced into two pages. You may find stage IV at the beginning of the next page.
We deeply appreciate the valuable time from reviewers and editors. Looking forward to knowing this manuscript is acceptable by the Journal.
Round 2
Reviewer 1 Report
My previous comments have been adequately addressed in the author's response